
# Low rank compression in the numerical solution of the nonequilibrium Dyson equation

**Jason Kaye[1,2] and Denis Golež[2,3,4]**

**1** Center for Computational Mathematics, Flatiron Institute,
New York, New York 10010, USA
**2** Center for Computational Quantum Physics, Flatiron Institute,
New York, New York 10010, USA
**3** Institute Jožef Stefan, Jamova 39, SI-1001 Ljubljana, Slovenia
**4** Department of Physics, Faculty of Mathematics and Physics,
University of Ljubljana, SI-1000 Ljubljana, Slovenia

## Abstract

We propose a method to improve the computational and memory efficiency of numerical solvers for the nonequilibrium Dyson equation in the Keldysh formalism. It is based on the empirical observation that the nonequilibrium Green's functions and self energies arising in many problems of physical interest, discretized as matrices, have low rank off-diagonal blocks, and can therefore be compressed using a hierarchical low rank data structure. We describe an efficient algorithm to build this compressed representation on the fly during the course of time stepping, and use the representation to reduce the cost of computing history integrals, which is the main computational bottleneck. For systems with the hierarchical low rank property, our method reduces the computational complexity of solving the nonequilibrium Dyson equation from cubic to near quadratic, and the memory complexity from quadratic to near linear. We demonstrate the full solver for the Falicov-Kimball model exposed to a rapid ramp and Floquet driving of system parameters, and are able to increase feasible propagation times substantially. We present examples with 262 144 time steps, which would require approximately five months of computing time and 2.2 TB of memory using the direct time stepping method, but can be completed in just over a day on a laptop with less than 4 GB of memory using our method. We also confirm the hierarchical low rank property for the driven Hubbard model in the weak coupling regime within the GW approximation, and in the strong coupling regime within dynamical mean-field theory.

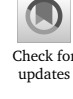

# 1 Introduction

The numerical solution of the quantum many-body problem out of equilibrium is an outstanding challenge in modern physics, required to simulate the effect of strong radiation fields on atoms and molecules [1,2], quantum materials [3–5], nuclear physics [6–8], ultracold atomic gases [9–11], and many other systems. Various theoretical frameworks for equilibrium problems have been extended to the nonequilibrium situation, including density functional theory [12], the density matrix renormalization group (DMRG) [13], and field theory approaches based on the Keldysh formalism [14–17]. A typical limitation is the restriction to rather short propagation times. This inherent difficulty manifests itself in various forms; for example, in bond dimension growth in DMRG [13], the dynamical sign problem in Monte Carlo methods [18–20], and memory effects in the Keldysh formalism [15,17,21,22]. Extending propagation times would allow for the investigation of new phenomena, such as the stabilization of metastable states [23–26], which take place on time scales that are orders of magnitude larger than those currently reachable by state-of-the-art techniques.

The Keldysh formalism is a particularly versatile approach, as it is not limited by the dimension of the problem (like DMRG), and can be efficiently adjusted to realistic setups. Several recent studies have used these techniques in direct comparisons with experiments, including those involving transport properties [27] and periodic driving [11] in ultra-cold atomic systems, and pump-probe experiments in correlated solids [5,28–32]. Many have already reached the level of first-principles description [33–37]. The essential task is to evaluate the two-time Green's function either with a numerically exact method [38–40] or with a high-order approximation scheme adjusted for the problem being studied [17,30,41–49].

Unfortunately, because of the necessity of computing full history integrals, the solution of the underlying Dyson equation by standard algorithms has a computational complexity of $\mathcal{O}(N^3)$ and a memory complexity of $\mathcal{O}(N^2)$ with respect to the number $N$ of time steps [15]. Numerous proposals have been made to improve efficiency, including memory truncation [50,

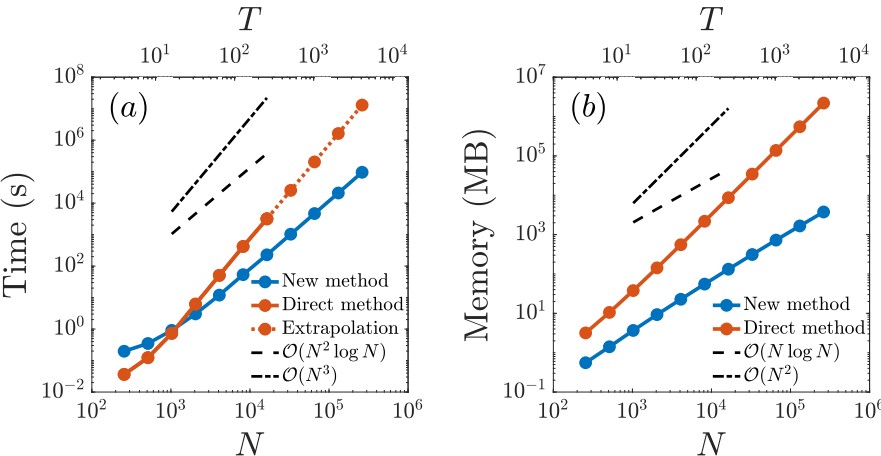

Figure 1: (a) Wall clock time and (b) memory used by the direct method and our method for the Falicov-Kimball model with a rapid ramp of the interaction parameter, showing the improved scalings. The time step $\Delta t$ is fixed, and the number of time steps is increased from $N = 256$ to $262\,144$, corresponding to an increase of the propagation time from $T = 4$ to $4096$. The direct method is impractical for large $N$, so we show extrapolated timings.

51], high-order time stepping and quadrature rules [22], parallelization [49, 52], and direct Monte Carlo sampling at long times [38, 40]. Of these, only the memory truncation methods succeed in systematically reducing the asymptotic cost and memory complexity, but these are restricted to specific parameter regimes in which the Green's functions are numerically sparse [50, 51]. Another alternative is to approximate the full propagation scheme with quantum kinetic equations or their generalizations, like the generalized Kadanoff-Baym ansatz (GKBA) [53–55]. These techniques are sufficiently accurate to explore long-time processes in several setups [22, 56–58]. However, they rely on an approximation for the equations of motion which is not always justified [59, 60]. Moreover, they lose their advantageous scaling for higher-order expansions.

There is still a demand, therefore, for a versatile propagation scheme with reduced computational and memory complexity which is compatible with recently developed nonperturbative techniques, like time-dependent dynamical mean-field theory (DMFT) [17, 41] and numerically exact Monte-Carlo approaches [18–20, 38–40]. In this work, we propose a method which, for systems whose Green's functions and self energies have the so-called hierarchical off-diagonal low rank (HODLR) property, reduces the computational complexity from $\mathcal{O}(N^3)$ to $\mathcal{O}(N^2 \log N)$ and the memory complexity from $\mathcal{O}(N^2)$ to $\mathcal{O}(N \log N)$. The HODLR structure allows us to build a compressed representation of Green's functions and self energies on the fly using the truncated singular value decomposition, and we use this representation to accelerate the evaluation of history integrals. We have confirmed the HODLR property in several systems of physical interest, and present results for the Falicov-Kimball model and the Hubbard model excited by a rapid ramp or a periodic driving. Our numerical examples show simulations with unprecedented propagation times, and computational cost and memory reductions of orders of magnitude. Scaling results for an example using the Falicov-Kimball model are shown in Fig. 1.

Our method may be integrated into existing time stepping schemes, including high-order discretizations, with user-controllable accuracy. That is, the additional error is set by a user-defined parameter $\varepsilon$, and the cost and memory requirements increase slowly as $\varepsilon$ is decreased for HODLR systems. Efficiency is achieved not by making additional modeling assumptions, but by exploiting an existing compressibility structure. Notably, the algorithm discovers the

ranks in the HODLR representation automatically, and if a system fails to obey the HODLR property to some degree, the algorithm will simply slow down accordingly in order to guarantee $\varepsilon$ accuracy, rather than give an incorrect result.

This article is organized as follows. In Sec. 2, we describe the Kadanoff-Baym form of the nonequilibrium Dyson equation and review the standard method of solving it. In Sec. 3, we introduce the HODLR compression structure, show how to build it on the fly, and describe a fast algorithm for the evaluation of history integrals. In Sec. 4, we demonstrate a full implementation for the Falicov-Kimball model, and study the HODLR compressibility of Green's functions for the Hubbard model excited by a rapid ramp and by periodic driving of system parameters. In Sec. 5, we summarize our results and discuss several future directions of research.

## 2  The Kadanoff-Baym equations

The Keldysh formalism describes the single particle two-time Green's functions

$$G_{jk}(t,t') = -i\langle T_{\mathcal{C}} \hat{c}_j(t)\hat{c}_k^\dagger(t')\rangle\,, \tag{1}$$

where $\hat{c}_j^\dagger$ ($\hat{c}_j$) denotes the fermionic or bosonic creation (annihilation) operator with respect to the $j^{\text{th}}$ single particle state, and $T_{\mathcal{C}}$ is the contour order operator; see Refs. [15, 22]. The construction of the Green's functions is typically carried out by first evaluating the self energy diagrams defining whichever approximation is employed, and then solving the Dyson equation, which resums those diagrams up to infinite order. In this work, we focus on the second step, and consider situations in which the solution of the Dyson equation is the computational bottleneck, although our method may still yield a significant reduction in memory usage in other cases.

The nonequilibrium Dyson equation in the integro-differential form is given by

$$(i\partial_t - h(t))\,G(t,t') - \int_{\mathcal{C}} d\bar{t}\,\Sigma(t,\bar{t})G(\bar{t},t') = \delta_{\mathcal{C}}(t,t'), \tag{2}$$

where $h$ is the single particle Hamiltonian and $\Sigma$ is the self energy. Here, for simplicity, we consider the scalar case $j = k = 1$, but the extension to the multidimensional case is straightforward. $\Sigma$ in general depends nonlinearly on $G$. The Green's function is typically parametrized in terms of Keldysh components, and for the solution of the Dyson equation, it is particularly useful to employ a set of physical components: the Matsubara component $G^M$ , the retarded component $G^R$, the left-mixing component $G^{\rceil}$, and the lesser component $G^<$. The equations of motion for these components lead to the *Kadanoff-Baym equations,* a set of causal coupled nonlinear Volterra integro-differential equations (VIDEs) given by [8, 15, 17, 21]

$$(-\partial_\tau - h(0))\,G^{\mathrm{M}}(\tau) - \int_0^\beta d\bar{\tau}\,\Sigma^{\mathrm{M}}(\tau - \bar{\tau})G^{\mathrm{M}}(\bar{\tau}) = 0 \tag{3}$$

$$\left(-\mathrm{i}\partial_{t'} - h(t')\right)G^R(t,t') - \int_{t'}^t d\bar{t}\,G^R(t,\bar{t})\Sigma^R(\bar{t},t') = 0 \tag{4}$$

$$(\mathrm{i}\partial_t - h(t))\,G^{\rceil}(t,\tau) - \int_0^t d\bar{t}\,\Sigma^{\mathrm{R}}(t,\bar{t})G^{\rceil}(\bar{t},\tau) = \int_0^\beta d\bar{\tau}\,\Sigma^{\rceil}(t,\bar{\tau})G^{\mathrm{M}}(\bar{\tau} - \tau) \tag{5}$$

$$(\mathrm{i}\partial_t - h(t))\,G^<(t,t') - \int_0^t d\bar{t}\,\Sigma^{\mathrm{R}}(t,\bar{t})G^<(\bar{t},t')$$

$$= \int_0^{t'} d\bar{t}\,\Sigma^<(t,\bar{t})G^A(\bar{t},t') - \mathrm{i}\int_0^\beta d\bar{\tau}\,\Sigma^{\rceil}(t,\bar{\tau})G_1^{\lceil}(\bar{\tau},t') \tag{6}$$

along with the conditions

$$G^{\mathrm{M}}(-\tau) = \xi G^{\mathrm{M}}(\beta - \tau) \tag{7}$$

$$G^{R}(t, t) = -i \tag{8}$$

$$G^{\rceil}(0, \tau) = iG^{\mathrm{M}}(-\tau) = i\xi G^{\mathrm{M}}(\beta - \tau) \tag{9}$$

$$G^{<}(0, t') = -\overline{G^{\rceil}(t', 0)} \tag{10}$$

and the relations

$$G^{\lceil}(\tau, t) = -\xi \overline{G^{\rceil}(t, \beta - \tau)} \tag{11}$$

$$G^{A}(t, t') = \overline{G^{R}(t', t)} \tag{12}$$

$$G^{<}(t, t') = -\overline{G^{<}(t', t)}. \tag{13}$$

Here $\xi = \pm 1$ for the bosonic and fermionic cases, respectively, $\overline{\cdot}$ denotes complex conjugation, $\beta$ is the given inverse temperature, and $\tau$ is an imaginary time variable. We note that we have used the conjugate equation for the retarded component in (4). For a detailed derivation of the Kadanoff-Baym equations, we refer the reader to Ref. [15].

## 2.1 Direct solution of the Kadanoff-Baym equations

Our method is built on top of the direct time stepping procedure which has traditionally been used to solve the Kadanoff-Baym equations. We now briefly review that procedure. For details about discretization, initialization procedures for high-order methods, nonlinear iteration, and the evaluation of self energies, we refer the reader to Refs. [15, 22].

We assume a discretization of the variables $t$ and $t'$ on a uniform grid $t_n = n\Delta t$ with $n = 0, 1, \ldots, N$, and of the variable $\tau$ on a uniform grid $\tau_k = k\Delta\tau$ with $k = 0, 1, \ldots, M$. The final time is given by $T = N\Delta t$, and the inverse temperature by $\beta = M\Delta\tau$. The Green's functions are sampled on the appropriate products of these grids to form matrices, except for the Matsubara component, which is only a function of $\tau$. The retarded Green's function is represented by a lower triangular matrix, and because of its Hermitian antisymmetry (13), the lesser Green's function is determined by its lower or upper triangular part [22].

First, Eqns. (3) and (7) for the Matsubara component may be solved independently of the other components. Several efficient numerical methods exist [22, 61, 62], and we do not consider this topic here.

The entries for each of the other Green's functions are computed in the following order:

- The lower triangular matrix $G^{R}(t_m, t_n)$ is filled in with $m$ proceeding from 0 to $N$ in an outer iteration, and with $n$ proceeding from $m$ to 0 in an inner iteration.

- The rectangular matrix $G^{\rceil}(t_m, \tau_k)$ is filled in with $m$ proceeding from 0 to $N$ and for each $k$ in parallel.

- The upper-triangular matrix $G^{<}(t_n, t_m)$ is filled in with $m$ proceeding from 0 to $N$ in an outer iteration, and with $n$ proceeding from 0 to $m$ in an inner iteration.

More specifically, suppose we have reached the outer time step $t_{m'} = m'\Delta t$, so that $G^{R}(t_m, t_n)$, $G^{\rceil}(t_m, \tau_k)$, and $G^{<}(t_n, t_m)$ are known for $m = 0, 1, \ldots, m'-1$, $n = 0, 1, \ldots, m$, and $k = 0, 1, \ldots, M$. We can then fill in the matrix entries corresponding to $m = m'$, $n = 0, 1, \ldots, m'$, and $k = 0, 1, \ldots, M$. Since the self energies depend in general on the values of the Green's functions at these points, we must carry out a self-consistent iteration on the new entries of the Green's functions. At the beginning of each iterate, the new entries of the self energies are first

computed based on some combination of extrapolation from previous time steps and previous iterates of the current time step. Then, assuming fixed self energies, the new entries of the Green's functions for a given iterate are computed by the following procedure:

1. The equation (4) for the retarded component is a linear VIDE in $t'$ for fixed $t = t_{m'}$. It is solved by a time stepping procedure, starting at $t' = t_{m'} = t$, where we use the condition (8), backwards to $t' = t_0 = 0$. The result is the new row $\{G^R(t_{m'}, t_n)\}_{n=0}^{m'}$.

2. The equation (5) for the left-mixing component is a system of coupled linear VIDEs indexed by $\tau = \tau_k$. It is solved for the new row $\{G^\rceil(t_{m'}, \tau_k)\}_{k=0}^{M}$. Note that at the first time step, we use the initial condition (9).

3. The equation (6) for the lesser component is a linear VIDE in $t$ for fixed $t' = t_{m'}$. It is solved starting at $t = t_0 = 0$, where we have the now known condition (10), forwards to $t = t_{m'} = t'$. The result is the new column $\{G^<(t_n, t_{m'})\}_{n=0}^{m'}$. Note that as a consequence of the Hermitian antisymmetry (13) of $G^<$ and the definitions (11) and (12), the right hand side of (6) is entirely known.

The most expensive step in the solution of each VIDE is the evaluation of the various integrals at each time step. We refer to these as history integrals, or history sums when discretized, since they involve summation over previously computed quantities. To understand the cost, we first consider the history integral in the VIDE for the retarded component corresponding to the outer time step $t = t_m$ and inner time step $t = t_n$, discretized by the trapezoidal rule as

$$I_{m,n}^{R,1} = \Delta t \sum_{j=n}^{m}{}' G^R(t_m, t_j) \Sigma^R(t_j, t_n). \tag{14}$$

Here, the primed sum symbol indicates that the first and last terms of the sum are weighted by $1/2$. The specific discretization used is unimportant for our discussion, and we have chosen the trapezoidal rule for simplicity. For each $m = 0, \dots, N$, we must compute this sum for $n = 0, \dots, m$, so that the cost of computing all such sums scales as $\mathcal{O}(N^3)$. By contrast, the cost of time stepping for all outer time steps $t_m$ and inner time steps $t_n$, ignoring the history sums, scales as $\mathcal{O}(N^2)$. Furthermore, computing these sums requires storing $\Sigma^R(t_m, t_n)$ in its entirety, an $\mathcal{O}(N^2)$ memory cost.

In Table 1, we list the six history sums, obtained by discretizing the corresponding integrals in Eqns. (4)–(6), along with the total cost of computing them directly. Each history sum is slightly different, but for each Keldysh component, the cost of computing the history sums is dominant. Furthermore, in order to compute the sums, one must store all of the computed Green's functions and/or the corresponding self energies. Our main objective is to reduce these bottlenecks.

Before discussing our approach, it will be useful to understand the history sums in terms of matrix algebra. We again use the retarded component as an example. At each outer time step $t_m$, the collection of sums $\{I_{m,n}^{R,1}\}_{n=0}^{m}$ may be viewed as the product of a $1 \times m$ row vector $\{G^R(t_m, t_j)\}_{j=0}^{m}$ with an $m \times m$ lower triangular matrix $\{\Sigma^R(t_j, t_n)\}_{j=0\ n=0}^{m\ \ j}$, properly modified to take the trapezoidal rule weights into account. The cost of computing each such product is $\mathcal{O}(m^2)$, so that the cost of computing all such products is $\mathcal{O}(N^3)$. Of course, we cannot compute all sums simultaneously by such a matrix-vector product, since $\{G^R(t_m, t_j)\}_{j=0}^{m}$ is itself built during the course of time stepping. Rather, this product is computed one step at a time; at the time step $t_n$, we compute the product of the row vector with the $n^{\text{th}}$ column of the lower triangular matrix.

Table 1: History sums and the total cost of computing them for all index values by direct summation and by our method. The sums are trapezoidal rule discretizations of the underlying integrals. $k$ is a bound on the ranks of all blocks in the compressed representations. The total cost of building these representations by on the fly TSVD updates is $\mathcal{O}\left(k^2\left(N^2 + NM\right)\right)$.

| History sum | Direct summation cost | Fast summation cost |
|---|---|---|
| $I_{m,n}^{R,1} = \Delta t \sum_{j=n}^{m}{}' G^R(t_m, t_j)\Sigma^R(t_j, t_n)$ | $\mathcal{O}\left(N^3\right)$ | $\mathcal{O}\left(kN^2 \log N + k^2 N^2\right)$ |
| $I_{m,k}^{\rceil,1} = \Delta t \sum_{j=0}^{m}{}' \Sigma^R(t_m, t_j)G^{\rceil}(t_j, \tau_k)$ | $\mathcal{O}\left(N^2 M\right)$ | $\mathcal{O}\left(kN^2 + kNM + k^2 N\right)$ |
| $I_{m,k}^{\rceil,2} = \Delta\tau \sum_{l=0}^{M}{}' \Sigma^{\rceil}(t_m, \tau_l)G^M(\tau_l - \tau_k)$ | $\mathcal{O}\left(NM^2\right)$ | $\mathcal{O}\left(NM \log M\right)$ |
| $I_{n,m}^{<,1} = \Delta t \sum_{j=0}^{n}{}' \Sigma^R(t_n, t_j)G^<(t_j, t_m)$ | $\mathcal{O}\left(N^3\right)$ | $\mathcal{O}\left(kN^2 \log N + k^2 N^2\right)$ |
| $I_{n,m}^{<,2} = \Delta t \sum_{j=0}^{m}{}' \Sigma^<(t_n, t_j)G^A(t_j, t_m)$ | $\mathcal{O}\left(N^3\right)$ | $\mathcal{O}\left(kN^2 \log N + k^2 N^2\right)$ |
| $I_{n,m}^{<,3} = \Delta\tau \sum_{l=0}^{M}{}' \Sigma^{\rceil}(t_n, \tau_l)G^{\lceil}(\tau_l, t_m)$ | $\mathcal{O}\left(N^2 M\right)$ | $\mathcal{O}\left(kN^2 + kNM + k^2 N\right)$ |

# 3 A fast, memory-efficient method based on hierarchical low rank compression

To reduce the bottlenecks associated with history storage and summation, we might first hope that the Green's functions and self energies display some sort of sparsity. For example, if the retarded Green's function decays rapidly in the off-diagonal direction, we do not need to store its entries with moduli smaller than some threshold, and we can ignore them in the history sums. Though this is sometimes the case [50, 51], decay in the Green's functions and self energies depends strongly on the parameter regime. On the other hand, we have found that the Green's functions and self energies for many systems of physical interest display a smoothness property which leads to *data sparsity*; they have numerically low rank off-diagonal blocks. This can be systematically exploited to achieve highly compressed representations which admit simple algorithms for fast matrix-vector multiplication.

We will first discuss the compressed storage format for the Green's functions and self energies. Then we will describe how to build these compressed representations on the fly, as new matrix entries are filled in. Finally, we will show that the compressed format leads naturally to a fast algorithm to compute the history sums.

## 3.1 Hierarchical low rank compression of Green's functions

Consider first the retarded Green's function, which is discretized as a lower triangular matrix. We partition the matrix into blocks by recursive subdivision, as in Figure 2. Each block may be described by its *level*; we say that the largest block is in level one, the two second largest blocks are in level two, and so on, with $L$ levels in total. We choose $L \sim \log_2 N$ so that each of the triangular blocks near the diagonal contains a small, constant number of entries. All blocks are square, and the blocks in a given level have dimensions approximately half of those in the previous level.

A matrix is said to have the *hierarchical off-diagonal low rank (HODLR) property with $\varepsilon$-*

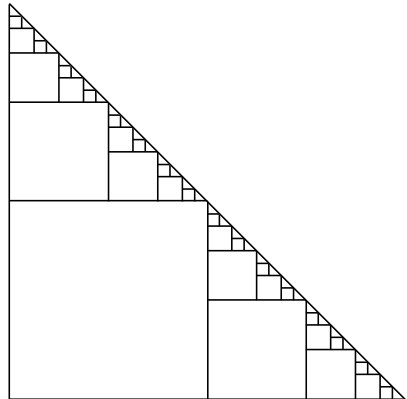

Figure 2: Hierarchical off-diagonal partioning of a lower triangular $N \times N$ matrix. The matrix is HODLR if the blocks in the recursive partitioning above, continued towards the diagonal until the smallest blocks are of a small constant size, have $\varepsilon$-rank $k \ll N$.

*rank $k$* if each of the blocks in this partitioning are rank $k$ to a threshold $\varepsilon$ – that is, their $(k+1)^{\text{th}}$ singular values are less than $\varepsilon$ [63]. Using the truncated singular value decomposition (TSVD), obtained from the SVD by setting all singular values less than $\varepsilon$ to zero and deleting the corresponding left and right singular vectors, an $n \times n$ block with $\varepsilon$-rank $k$ may be stored using $k(2n+1)$ rather than $n^2$ numbers, and recovered with error at most $\varepsilon$ in the spectral norm. The TSVD is the best rank $k$ approximation in this norm, with error given by the $(k+1)^{\text{th}}$ singular value [63, Sec. 2, Thm. 2]. Since each of the $2^{l-1}$ blocks in level $l$ has dimensions $n \approx N/2^l$, all blocks in a HODLR matrix may be stored with arbitrary accuracy $\varepsilon$ using approximately

$$k \sum_{l=1}^{L} 2^{l-1}(N/2^{l-1} + 1) = \mathcal{O}\left(kN \log_2 N\right)$$

rather than $\mathcal{O}\left(N^2\right)$ numbers. The entries in the triangular blocks near the diagonal may be stored directly, and since we choose $L \sim \log_2 N$, there are only $\mathcal{O}(N)$ of them.

We are primarily interested in the behavior of the family of matrices obtained by fixing $\Delta t$ and increasing $N$ to reach longer propagation times. It may be that in this family, the $\varepsilon$-rank bound $k$ itself increases with $N$. If $k$ grows linearly with $N$, then there is no asymptotic advantage to storing the matrix in the compressed format described above. If $k$ does not grow with $N$ at all – the ideal case – then the total cost of storage in the compressed format grows only as $\mathcal{O}(N \log N)$. We have examined the $\varepsilon$-rank behavior of retarded Green's functions and self energies in the compressed format for a variety of physical systems. *Our crucial empirical observations are, first, that for fixed $N$, the maximum $\varepsilon$-rank $k$ of any block is often much less than $N$, and second, that in these cases $k$ grows only weakly with $N$, so that storage costs are close to $\mathcal{O}(N \log N)$ with a small scaling constant.*

The matrices of the lesser Green's function and self energy are Hermitian antisymmetric, so we need only store their lower triangular parts. We have observed that they have similar $\varepsilon$-rank structures to the retarded components. The left-mixing Green's function is represented by a full $N \times M$ matrix, and our observation is that this matrix is often simply of low $\varepsilon$-rank. Using the TSVD, then, we can store it using $k(N + M + 1)$ rather than $NM$ numbers, where here and going forward we use $k$ to denote an $\varepsilon$-rank bound for all Keldysh components.

We note that if $T$ is fixed and $N$ is increased in order to achieve higher resolution – the $\Delta t \to 0$ regime – the $\varepsilon$-ranks cannot increase asymptotically with $N$, and we are guaranteed $\mathcal{O}(N \log N)$ scaling of the memory usage. In this case, the size of the constant $k$ in the scaling entirely determines the efficiency of the method.

The rank properties, and consequently the compressibility, of the Green's functions and self energies vary from system to system. We give numerical evidence of significant compressibility for several systems of physical interest in Sec. 4.

## 3.2 Building the compressed representation on the fly

If we are required to construct each block in its entirety before compressing it, the peak memory usage will still scale as $\mathcal{O}(N^2)$. Therefore, we must build the Green's functions and self energies in the compressed format on the fly. During the course of time stepping, new rows of the retarded Green's function matrix are filled in one at a time, according to the procedure described in Sec. 2.1. Each new row may be divided among the blocks in the HODLR partition containing a part of it. The new entries in the triangular blocks are stored directly. For the other blocks, we must compute the TSVD of the concatenation of a block known in TSVD representation with a new row. We carry out this process, called an SVD update, using the method described in Ref. [64, Secs. 2-3], which we outline below.

Suppose we have a matrix $B'$ given by the first $m'$ rows of an $m \times n$ block $B$, with rank $k'$ TSVD $B' = U'S'V'^*$. Given a new row $b^*$ of $B$, we wish to construct the $\varepsilon$-rank TSVD of the matrix

$$B^+ = \begin{pmatrix} B' \\ \hline b^* \end{pmatrix}$$

in an efficient manner – in particular, we cannot simply expand $B'$ from its TSVD, append $b^*$, and compute the TSVD of $B^+$ directly, since this would lead to $\mathcal{O}(N^2)$ peak memory usage. If $m' = 0$, $B'$ is empty, and the TSVD is simply given by $B^+ = U^+S^+V^{+*}$ with $U^+ = 1$, $S^+ = 1$, and $V^+ = b$. Otherwise, we begin by writing

$$B^+ = \left( \begin{array}{c|c} U' & 0 \\ \hline 0 & 1 \end{array} \right) \left( \begin{array}{c|c} S' & 0 \\ \hline 0 & 1 \end{array} \right) \left( \begin{array}{c} V'^* \\ \hline b^* \end{array} \right).$$

We first orthogonalize the new row against the current row space of $B$, which is given by the span of the columns of $V'$. In particular, define the normalized orthogonal complement $q = (b - V'V'^*b)/\beta$ with $\beta = \|b - V'V'^*b\|$. $V'^*b$ and $q$ may be computed at a cost of $\mathcal{O}(nk')$ using the modified Gram-Schmidt algorithm [65, Sec. 5.2.8]. We then have

$$\left( \begin{array}{c|c} V' & b \end{array} \right) = \left( \begin{array}{c|c} V' & q \end{array} \right) \left( \begin{array}{c|c} I & V'^*b \\ \hline 0 & \beta \end{array} \right)$$

and therefore

$$B^+ = \left( \begin{array}{c|c} U' & 0 \\ \hline 0 & 1 \end{array} \right) \left( \begin{array}{c|c} S' & 0 \\ \hline b^*V' & \beta \end{array} \right) \left( \begin{array}{c} V'^* \\ \hline q^* \end{array} \right). \tag{15}$$

The middle matrix is a $k'+1 \times k'+1$ half-arrowhead matrix, and we can compute its SVD $USV^*$ in $\mathcal{O}(k'^2)$ time [66]. The rank $k'+1$ TSVD of $B^+$ is then given by $B^+ = U^+S^+V^{+*}$, with $U^+ = \left( \begin{array}{c|c} U' & 0 \\ \hline 0 & 1 \end{array} \right) U$, $S^+ = S$, and $V^+ = \left( \begin{array}{c|c} V' & q \end{array} \right) V$. The cost of forming $U^+$ and $V^+$ by matrix-matrix multiplication is $\mathcal{O}(k'^2(m'+n))$, and is the asymptotically dominant cost in the update. If $S^+_{k'+1,k'+1} < \varepsilon$, then $B^+$ has $\varepsilon$-rank $k'$, and we remove the last column of $U^+$, the last row of $V^+$, and the last row and column of $S^+$.

The cost of building a full $m \times n$ block of rank at most $k$ one row at a time in this manner is $\mathcal{O}(k^2 m(m+n))$. For the retarded and lesser Green's functions and self energies, in each block we have $m \approx n$, with $2^{l-1}$ blocks at level $l$ of dimensions $n \approx N/2^l$, so the total update cost is of the order

$$k^2 \sum_{l=1}^{L} 2^{l-1} (N^2/2^{2l}) = \mathcal{O}(k^2 N^2).$$

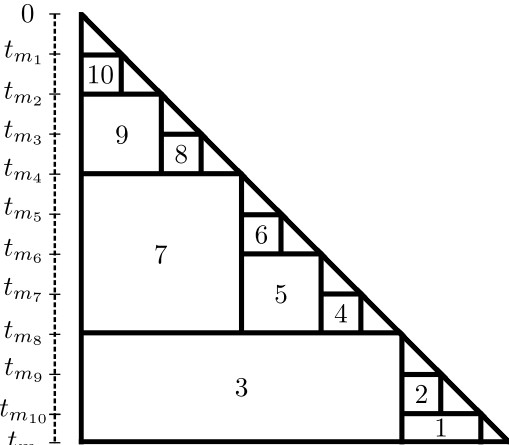

Figure 3: The portion of compressed representation depicted in Fig. 2 which has been constructed by the outer time step $t = t_m$ for the retarded Green's function and self energy. The block numbering indicates the order in which blocks are applied when computing the history sums $I^{R,1}_{n,m}$.

The left-mixing Green's function and self energy are $N \times M$ matrices with rows filled in one by one, so we can use the update procedure for a single block, at a total cost of $\mathcal{O}\left(k^2\left(N^2 + NM\right)\right)$.

## 3.3 Fast evaluation of history sums for the retarded component

Fig. 3 depicts the portion of the compressed representation of the retarded Green's function and self energy that has been built by the time we have reached the outer time step $t = t_m$. For the self energy, this is the compressed representation of the integral kernel appearing in the VIDE (4) corresponding to $t = t_m$. We describe in this section how to use this compressed representation to efficiently compute the history sums $I^{R,1}_{n,m}$ defined in (14), for $n = 0, \dots, m$.

On the left side of Fig. 3, we show a division of the interval $[0, t_m]$ into panels with endpoints $0, t_{m_1}, t_{m_2}, \dots, t_{m_{10}}, t_m$. $m_1, m_2, \dots, m_{10}$ are the first row indices of the 10 blocks, ordered from top to bottom. As described in Sec. 2.1, when solving the VIDE corresponding to $t = t_m$ by time stepping, the row vector $\{G^R(t_m, t_j)\}_{j=0}^m$ is filled in from right to left; we start with $G^R(t_m, t_m) = -i$, then we fill in $G^R(t_m, t_{m-1})$, and so on, until we reach $G^R(t_m, 0)$. As shown in Eq. (14), the history sum $I^{R,1}_{m,n}$ corresponding to time step $t' = t_n$ is given by the weighted product of the row vector $\{G^R(t_m, t_j)\}_{j=n}^m$ with the column vector $\{\Sigma^R(t_j, t_n)\}_{j=n}^m$.

From time steps $t_m$ until $t_{m_{10}}$, the portion of the kernel matrix contributing to the history sums is the lower right triangular block, and we compute the sums directly. However, once we fill in $G^R(t_m, t_{m_{10}})$, we can apply the block labeled by 1 to the row vector $\{G^R(t_m, t_j)\}_{j=m_{10}}^m$, yielding a partial contribution to the history sums $I^{R,1}_{m,n}$ with $n$ indexing the columns of that block. These contributions are stored. Once we have filled in $G^R(t_m, t_{m_9})$, we can apply the block labeled by 2 to the row vector $\{G^R(t_m, t_j)\}_{j=m_9}^{m_{10}}$, yielding partial contributions to the history sums corresponding to its columns, which are added to the previous contributions and stored. Once we have filled in $G^R(t_m, t_{m_8})$, we can apply the block labeled by 3 to the row vector $\{G^R(t_m, t_j)\}_{j=m_8}^m$, again obtaining part of the corresponding history sums. We proceed in this manner, applying blocks as soon as all entries of $\{G^R(t_m, t_j)\}_{j=0}^m$ corresponding to the block row indices become available, and adding the result to the history sums corresponding to the block column indices.

By the time we reach some time step $t_n$, we will have already computed $I^{R,1}_{m,n}$, except for the *local part* of the sum; that is, the product of the part of the $n$th column contained in a triangular

block with the corresponding part of $\{G^R(t_m, t_j)\}_{j=0}^m$, which contains the most recently added entries. We compute this small dot product directly and add it in to obtain the full history sum $I_{m,n}^{R,1}$.

So far, we have not described a fast algorithm, only a way of reorganizing the computation of the history sums into a collection of matrix-vector products and small dot products. To obtain a fast algorithm, we recall that each block $B$ is stored as a TSVD $B = USV^*$, with the $\varepsilon$-rank bound $k$. If $B$ is $m \times n$, then it can be applied with a cost scaling as $\mathcal{O}(k(m + n + k))$ rather than $\mathcal{O}(mn)$ by applying the factors of the TSVD one at a time. Suppose then that the blocks, including partial blocks, in the compressed representation of the kernel at the current outer time step $t = t_m$ are in levels at least $l'$. Since there are $\sim 2^l m/N$ blocks each of dimensions $N/2^l$ at level $l$, the cost of applying all blocks at that time step is bounded asymptotically by

$$k \sum_{l'}^{L} \frac{m}{N} 2^l \left(N/2^l + k\right) = km(L - l' + 1) + \frac{2mk^2}{N}\left(2^{L+1} - 2^{l'}\right).$$

Since there are approximately $N/2^{l'-1}$ time steps $t_m$ for which the blocks are in levels at least $l'$, and $m \le N/2^{l'-1}$ for such a time step, the total cost of applying all blocks at every time step is therefore

$$k \sum_{l'=1}^{L} \frac{N}{2^{l'-1}} \left( \frac{N}{2^{l'-1}}(L - l' + 1) + \frac{2k}{N} \frac{N}{2^{l'-1}} \left(2^{L+1} - 2^{l'}\right) \right) = \mathcal{O}\left(kN^2 \log_2 N + k^2 N^2\right).$$

The total cost of computing the local sums is only $\mathcal{O}(N^2)$.

## 3.4 Fast evaluation of history sums for the other components

Each of the history sums defined in Table 1 is evaluated by a slightly different algorithm, but using similar ideas.

### 3.4.1 The sum $I_{m,k}^{\rceil,1}$

At the outer time step $t_m$, we compute the product of the row vector $\{\Sigma^R(t_m, t_j)\}_{j=0}^m$ with the rectangular matrix $\{G^{\rceil}(t_j, \tau_k)\}_{j=0\ k=0}^{m\ \ M}$, which is stored as a TSVD, to obtain $I_{m,k}^{\rceil,1}$ for $k = 0, \ldots, M$. Applying the factors of the TSVD one by one, we can carry out this product in $\mathcal{O}(k(m + M + k))$ operations, which gives a total cost for all sums scaling as $\mathcal{O}(kN^2 + kNM + k^2N)$.

### 3.4.2 The sum $I_{m,k}^{\rceil,2}$

We compute the product of the row vector $\{\Sigma^{\rceil}(t_m, \tau_l)\}_{l=0}^M$ with the square matrix $\{G^M(\tau_l - \tau_k)\}_{l=0\ k=0}^{M\ \ M}$ to obtain $I_{m,k}^{\rceil,2}$ for $k = 0, \ldots, M$. Here, we make use of a different sort of fast algorithm. $\{G^M(\tau_l - \tau_k)\}_{l=0\ k=0}^{M\ \ M}$ is an $M \times M$ Toeplitz matrix, and therefore can be applied in $\mathcal{O}(M \log M)$ time using the fast Fourier transform (FFT) [65, Sec. 4.7.7]. Briefly, this algorithm works by embedding the Toeplitz matrix in a larger circulant matrix, zero-padding the input vector, conjugating the circulant matrix by the discrete Fourier transform (DFT), which diagonalizes it, and applying the DFT and its inverse using the FFT. Using this algorithm gives a total cost of $\mathcal{O}(NM \log M)$ for all the sums.

### 3.4.3 The sum $I_{n,m}^{<,1}$

The sums $I_{n,m}^{<,1}$ for $n = 0, 1, \ldots, m$ are given by the product of the lower triangular matrix $\{\Sigma^R(t_n, t_j)\}_{n=0\ j=0}^{m\ \ n}$, stored in the compressed representation, with the column vector

$\{G^<(t_j, t_m)\}_{j=0}^n$. As for the retarded case, this column vector is filled in one entry at a time during the course of time stepping, from $j = 0$ to $j = m$. The algorithm to compute these sums on the fly is then analogous to that for the retarded case, except that the blocks are applied in order of increasing column index rather than decreasing row index as depicted in Fig. 3. The asymptotic cost is the same as for $I_{n,m}^{R,1}$.

### 3.4.4 The sum $I_{n,m}^{<,2}$

We compute the product of the square matrix $\{\Sigma^<(t_n, t_j)\}_{n=0}^m {}_{j=0}^m$ with the column vector $\{G^A(t_j, t_m)\}_{j=0}^m$ to obtain $I_{n,m}^{<,2}$ for $n = 0, \dots, m$. We have $\{G^A(t_j, t_m)\}_{j=0}^m = \{\overline{G^R(t_m, t_j)}\}_{j=0}^m$, which is known, and $\{\Sigma^<(t_n, t_j)\}_{n=0}^m {}_{j=0}^m$ is Hermitian antisymmetric, and therefore also fully known in the compressed representation. The same procedure as in Sec. 3.4.3 can be used to apply each block of the lower triangular part, and the upper triangular part can be applied simultaneously using the anti-conjugate transposes of the TSVDs of each block. The total asymptotic cost is therefore again the same as for $I_{n,m}^{R,1}$.

### 3.4.5 The sum $I_{n,m}^{<,3}$

We compute the product of the matrix $\{\Sigma^\rceil(t_n, \tau_l)\}_{n=0}^m {}_{l=0}^M$ with the column vector $\{G^\lceil(\tau_l, t_m)\}_{l=0}^M$ to obtain $I_{n,m}^{<,3}$ for $n = 0, \dots, m$. We perform the matrix-vector product using the TSVD of $\{\Sigma^\rceil(t_n, \tau_l)\}_{n=0}^m {}_{l=0}^M$, at a cost of $\mathcal{O}(k(m + M + k))$, giving a total cost of $\mathcal{O}(kN^2 + kNM + k^2N)$ for all sums.

## 3.5 Summary of the time stepping algorithm

The tools we have described can be integrated into the direct solution method discussed in Sec. 2.1 with only a couple of modifications:

- At the end of self-consistent iteration for each outer time step, the TSVD update algorithm described in Sec. 3.2 must be used to add the new rows of all Green's functions and self energies to their compressed representations.

- The fast procedures described in Secs. 3.3 and 3.4 must be used to compute all history sums.

By the end of the procedure, we will have computed compressed representations of each of the Green's functions and self energies. Operations may be carried out by working directly with the compressed representations, as in Secs. 3.3 and 3.4. An entry of a matrix stored in compressed format may be recovered in $\mathcal{O}(k)$ operations.

The costs associated with computing the history sums and updating the compressed representations are summarized in Table 1 and its caption. The storage costs scale as $\mathcal{O}(k(N \log N + M))$.

# 4 Numerical results

In this section, we demonstrate a full implementation of our method for a driven Falicov-Kimball model in the DMFT limit, using two nonequilibrium protocols: a fast ramp and periodic driving. We also test the efficiency of HODLR compression offline for the weak and strong coupling regimes of the Hubbard model. The weak coupling regime is described within the time-dependent GW approximation for a one dimensional system. The strong coupling (Mott

insulating) regime is described within the DMFT approximation on the Bethe lattice with a non-crossing approximation (NCA) as the impurity solver.

## 4.1 Full implementation for the Falicov-Kimball model

The Falicov-Kimball problem [67,68] describes a lattice consisting of itinerant $c$ electrons and immobile $f$ electrons which interact via a repulsive Coulomb potential with strength U. The Hamiltonian is given by

$$H = -J \sum_{\langle i,j \rangle} c_i^\dagger c_j + \epsilon \sum_i f_i^\dagger f_i + U \sum_i f_i^\dagger f_i c_i^\dagger c_i, \tag{16}$$

where we measure energies in the units of the hopping parameter $J$ and $\epsilon$ is the on-site energy. In the DMFT limit, the equilibrium phase diagram includes metallic, insulating, and charge density wave phases [69]. The effective local action for the itinerant electrons is quadratic and the problem can be solved numerically exactly [70,71], so the main computational bottleneck is the solution of the Dyson equation [72,73].

For simplicity, we will assume the Bethe lattice at half-filling, for which we obtain a pair of coupled equations of the form (3)–(13) for Green's functions $G_1$ and $G_2$, with

$$h_1(t) = U(t)/2, \quad h_2(t) = -U(t)/2,$$

and $\Sigma$ replaced by

$$\Delta(t,t') = \big(G_1(t,t') + G_2(t,t')\big)/2$$

for each Keldysh component of $G$. $\Delta$ is the hybridization function which enters the solution of the Kadanoff-Baym equations by replacing $\Sigma$, but is not equal to the self energy for the Falicov-Kimball system in DMFT. The Green's functions $G_1$ and $G_2$ correspond to the full or empty $f$ states after integrating out the $f$ electrons. In the Falicov-Kimball model on the Bethe lattice, the self consistency for the hybridization function $\Delta$ is simply a linear combination of $G_1$ and $G_2$, so no nonlinear iteration is required in its numerical solution. This is a convenient simplification, but does not materially affect our algorithm.

In the first example, we consider a rapid ramp of the interaction parameter $U(t)$, given by

$$U(t) = \frac{U_0 + U_1}{2} + \frac{U_1 - U_0}{2} \operatorname{erf}(5.922(2t - 1)). \tag{17}$$

$U$ starts in the metallic phase $U_0 = 1$ at inverse temperature $\beta = 5$, and smoothly increases deep into the insulator transition $U_1 = 8$. Experiments for various choices of $U_0$ and $U_1$ confirm that the significant compressibility of the solution which we will demonstrate for this case is typical.

In the second example, which we refer to as the Floquet example, we consider a periodic driving of system parameters. Such protocols have been studied extensively in the setting of Floquet engineering [74–77], Floquet prethermalization [45,78–80], and high-harmonic generation [11,81,82]. In particular, we simulate periodic driving of the Coulomb interaction,

$$U(t) = U_{\mathrm{eq}} + U_{\mathrm{dr}} \sin(\omega t), \tag{18}$$

where $U_{\mathrm{eq}}$ is the equilibrium interaction, $U_{\mathrm{dr}}$ is the driving strength, and $\omega$ is the driving frequency. We start in the insulating phase $U_{\mathrm{eq}} = 8$ at the inverse temperature $\beta = 5$ and choose a resonant excitation $\omega = U_{\mathrm{eq}}$ with strength $U_{\mathrm{dr}} = 2$. As in the ramp example, our experiments show that other parameter choices yield similar compressibility results. Plots of $G_1^R(t,t')$ for the two examples with propagation time $T = 8$ are shown in Fig. 4.

We march the integro-differential equations in time using the implicit trapezoidal rule, with history sums also discretized using the trapezoidal rule as in Sec. 2.1. It is not our intention

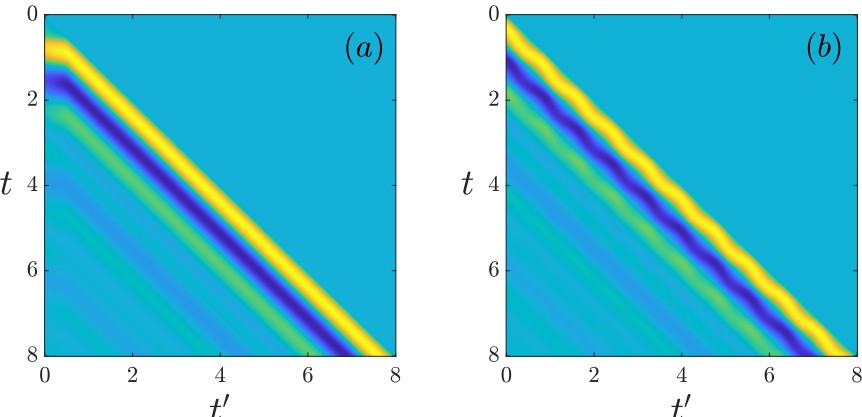

Figure 4: Plots of $G_1^R(t, t')$ for the (a) ramp and (b) Floquet examples of the Falicov-Kimball model, with propagation time $T = 8$. Note that the $t$ axis has been reversed to reflect the matrix representation which we work with in the previous sections.

here to obtain high accuracy calculations, and we use a low-order discretization for simplicity, but extension to high-order time stepping methods and quadrature rules for history summation is straightforward [22]. $G^M$ is computed in advance to near machine precision using a Legendre polynomial–based spectral method with fixed point iteration, and then evaluated on the equispaced $\tau$ grid [61, 62]. Codes were written in Fortan using OpenBLAS for matrix operations and linear algebra, including the evaluation of history sums in the direct method, and FFTW was used for the FFTs in the fast evaluation of the sums $I_{m,k}^{],2}$ [83–85]. All numerical experiments were performed on a laptop with an Intel Xeon E-2176M 2.7GHz processor.

In the following experiments, errors are always computed as the maximum entrywise difference between a computed solution and a reference. We note that this is not the norm in which the TSVD guarantees accuracy $\varepsilon$, but that is a minor technical issue and does not prevent effective error control. When we present $\varepsilon$-ranks associated with $G^R$ or $G^<$, we always maximize the rank over all blocks in the HODLR partition, and simply refer to this as the rank of the compressed representation. In all cases presented, for $G^R$ and $G^<$, this coincides with the $\varepsilon$-rank of the largest block in the partition. The number $L$ of levels can be adjusted to balance computational cost and memory usage, but for all experiments, we simply choose it so that the smallest blocks in the HODLR partition are approximately $16 \times 16$.

We first examine the behavior of our method as the SVD truncation tolerance $\varepsilon$ is varied, using both examples with the propagation time $T = 64$, corresponding to $N = 4096$ time steps, and $M = 128$ Matsubara time steps. Errors compared with the direct method, maximized over all Keldysh components of the Green's function, are given for several values of $\varepsilon$ in Table 2. For each experiment, the error is less than $\varepsilon$. In Fig. 5a, we plot the singular values of the largest blocks in the HODLR partitions of $G_1^R$ and $G_1^<$, and of $G_1^]$, for the ramp example. The singular values decay approximately exponentially, and as a result, the $\varepsilon$-ranks of these blocks increase only as approximately $\log(1/\varepsilon)$, as shown in Fig. 5b. The wall clock time required to compute each component of $G$, shown in Fig. 5c, increases slightly more slowly than the expected asymptotic $k \sim \log(1/\varepsilon)$ rate for these parameters. The memory required to store each component of $G_1$ is shown in Fig. 5d, and reflects the variation in ranks seen in Fig. 5b. The results for $G_2$ are similar. Fig. 6 contains analogous results for the Floquet example. Here, the decay of the singular values, while still rapid, is slightly slower than exponential, and this is reflected in the ranks, timings, and memory usage.

We next fix $T = 8$ and $\varepsilon = 10^{-4}$, and measure errors and ranks for $\Delta t$ corresponding

Table 2: Error of the Green's functions compared with the direct method, maximized over all Keldysh components, for different $\varepsilon$. For both examples, T = 64, N = 4096, and M = 128.

| $\varepsilon$ | | $10^{-2}$ | $10^{-4}$ | $10^{-6}$ | $10^{-8}$ | $10^{-10}$ |
|---|---|---|---|---|---|---|
| Error | Ramp | $6 \times 10^{-3}$ | $6 \times 10^{-5}$ | $6 \times 10^{-7}$ | $5 \times 10^{-9}$ | $5 \times 10^{-11}$ |
| | Floquet | $8 \times 10^{-3}$ | $6 \times 10^{-5}$ | $8 \times 10^{-7}$ | $7 \times 10^{-9}$ | $6 \times 10^{-11}$ |

to $N = 64, 128, \ldots, 8192$ time steps. $M$ is taken to be sufficiently large to eliminate it as a dominant source of error. The errors are measured against a well-resolved solution. The results are given in Table 3. We observe the expected second-order convergence with $\Delta t$, until the SVD truncation error is reached. We also find that the ranks are nearly constant as $N$ is increased. Indeed, once the solution is resolved by the grid, the block ranks cannot increase significantly as $\Delta t$ is further refined. In the regime of fixed $T$ and increasing $N$, therefore, we are guaranteed $\mathcal{O}\left(N^2 \log N\right)$ scaling of the computational cost and $\mathcal{O}(N \log N)$ scaling of the memory usage.

The more challenging regime is that of increasing $T$ and $N$ with fixed time step $\Delta t$. We take $\Delta t = 1/64$, and doubling values of the propagation time $T = 4, 8, \ldots, 4096$, corresponding to $N = 256, 512, 1024, \ldots, 262\,144$ time steps. We note that in our experiments, the maximum error for fixed $\Delta t$ is observed to be approximately constant as $N$ and $T$ are increased, so according to Table 3 these simulations have approximately three-digit accuracy. We fix $M = 128$, which is sufficient to eliminate it as a dominant source of error, and $\varepsilon = 10^{-4}$.

Table 3: Errors and ranks with a varying time step $\Delta t$ and fixed propagation time $T = 8$, for the ramp (top) and Floquet (bottom) examples. Errors and ranks for each component are maximized over $G_1$ and $G_2$.

| $\Delta t$ | Errors | | | Ranks | | |
|---|---|---|---|---|---|---|
| | $G^R$ | $G^{\rceil}$ | $G^{<}$ | $G^R$ | $G^{\rceil}$ | $G^{<}$ |
| 1/8 | $1.01 \times 10^{-1}$ | $5.06 \times 10^{-2}$ | $8.83 \times 10^{-2}$ | 9 | 7 | 8 |
| 1/16 | $2.50 \times 10^{-2}$ | $1.32 \times 10^{-2}$ | $2.25 \times 10^{-2}$ | 9 | 7 | 9 |
| 1/32 | $6.25 \times 10^{-3}$ | $3.32 \times 10^{-3}$ | $5.65 \times 10^{-3}$ | 9 | 7 | 9 |
| 1/64 | $1.56 \times 10^{-3}$ | $8.29 \times 10^{-4}$ | $1.42 \times 10^{-3}$ | 9 | 7 | 9 |
| 1/128 | $3.90 \times 10^{-4}$ | $2.05 \times 10^{-4}$ | $3.51 \times 10^{-4}$ | 10 | 7 | 9 |
| 1/256 | $9.78 \times 10^{-5}$ | $4.95 \times 10^{-5}$ | $8.56 \times 10^{-5}$ | 10 | 7 | 9 |
| 1/512 | $3.31 \times 10^{-5}$ | $4.79 \times 10^{-5}$ | $6.03 \times 10^{-5}$ | 10 | 7 | 9 |
| 1/1024 | $3.61 \times 10^{-5}$ | $4.79 \times 10^{-5}$ | $5.09 \times 10^{-5}$ | 11 | 7 | 10 |

| $\Delta t$ | Errors | | | Ranks | | |
|---|---|---|---|---|---|---|
| | $G^R$ | $G^{\rceil}$ | $G^{<}$ | $G^R$ | $G^{\rceil}$ | $G^{<}$ |
| 1/8 | $1.09 \times 10^{-1}$ | $1.07 \times 10^{-1}$ | $1.07 \times 10^{-1}$ | 9 | 6 | 8 |
| 1/16 | $2.81 \times 10^{-2}$ | $2.79 \times 10^{-2}$ | $2.79 \times 10^{-2}$ | 9 | 6 | 7 |
| 1/32 | $7.08 \times 10^{-3}$ | $7.02 \times 10^{-3}$ | $7.03 \times 10^{-3}$ | 9 | 6 | 8 |
| 1/64 | $1.78 \times 10^{-3}$ | $1.76 \times 10^{-3}$ | $1.77 \times 10^{-3}$ | 10 | 6 | 9 |
| 1/128 | $4.50 \times 10^{-4}$ | $4.40 \times 10^{-4}$ | $4.45 \times 10^{-4}$ | 10 | 6 | 9 |
| 1/256 | $1.22 \times 10^{-4}$ | $1.11 \times 10^{-4}$ | $1.20 \times 10^{-4}$ | 10 | 6 | 9 |
| 1/512 | $3.74 \times 10^{-5}$ | $5.65 \times 10^{-5}$ | $5.65 \times 10^{-5}$ | 10 | 6 | 9 |
| 1/1024 | $3.56 \times 10^{-5}$ | $5.94 \times 10^{-5}$ | $5.94 \times 10^{-5}$ | 11 | 6 | 9 |

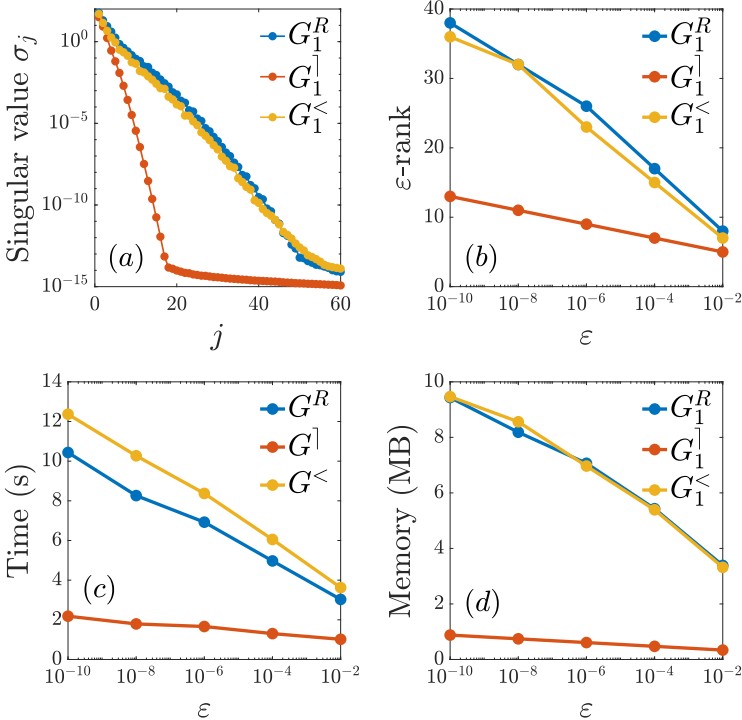

Figure 5: Varying the SVD truncation tolerance $\varepsilon$ for the ramp example with propagation time $T = 64$, corresponding to $N = 4096$ time steps. We plot (a) singular values of the largest block in the hierarchical partition for $G_1^R$ and $G_1^<$, and of $G_1^{\rceil}$, (b) the ranks of the compressed representations for $G_1$, (c) the time to compute each component of $G$, and (d) the memory required to store each component of $G_1$ in compressed form.

Fig. 1 shows the time and memory required for each simulation for the ramp example, using our algorithm and the direct method. For sufficiently large values of $N$, the direct method becomes impractical, and we obtain timings by extrapolation. We observe the expected scalings. For the largest simulation, which has $T = 4096$ and $N = 262\,144$, our algorithm takes approximately 26.5 hours and uses 3.8 GB of memory to store the Green's functions, whereas the direct method would take approximately 5 months and use 2.2 TB of memory. This implies a speedup factor of approximately 135, and a compression factor of approximately 580. A simulation which would take the direct method 24 hours, at $N \approx 49250$, using 78 GB of memory, would take our method approximately 42 minutes and use 512 MB of memory. Our method is faster whenever $N > 1200$, and it uses less memory for all values of $N$. The results for the Floquet example, given in Fig. 7, are nearly identical.

Rank information for the two examples is given in Figs. 8a and 8c, respectively. The crucial empirical observation enabling our complexity gains is that the maximum ranks grow at most logarithmically with $N$.

## 4.2 Hierarchical low rank compression in other systems

We have demonstrated an implementation specialized for the Falicov-Kimball problem for simplicity, but our method will be efficient for any system in which the Green's functions and self energies are HODLR compressible. This property can be tested offline, for a solution which has already been computed, by simply measuring the $\varepsilon$-ranks of the blocks in the compressed representations. In this way, we can determine whether or not our algorithm will be effective

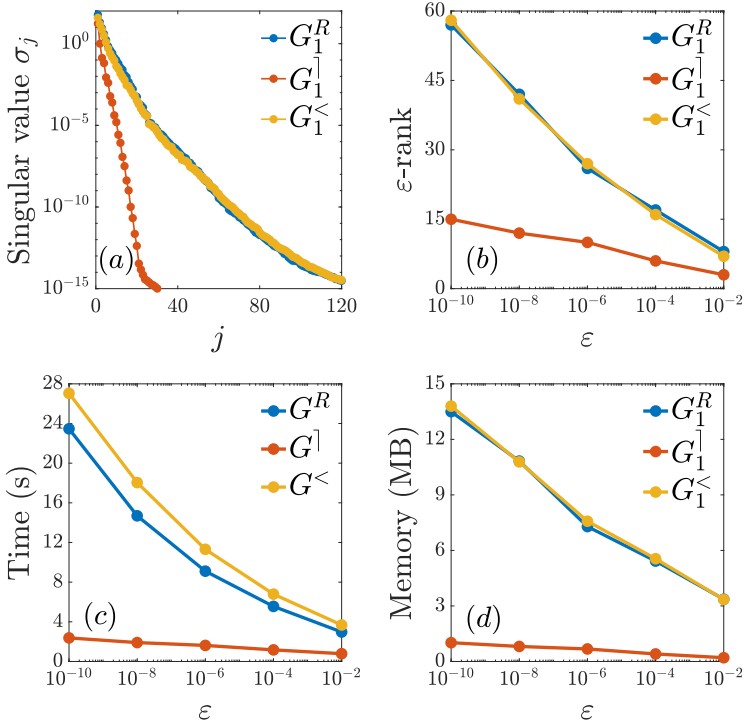

Figure 6: Varying the SVD truncation tolerance $\varepsilon$ for the Floquet example with propagation time $T = 64$, corresponding to $N = 4096$ time steps. The plots are analogous to those shown for the ramp example in Fig. 5.

when applied to other systems of interest.

Let us first verify for the Falicov-Kimball model that an offline measurement gives similar $\varepsilon$-ranks to those computed online using our SVD update algorithm. This will show that the update algorithm is yielding accurate results without requiring ranks which are larger than necessary, and therefore that measuring $\varepsilon$-ranks offline is sufficient to understand online performance. For both the ramp and Floquet examples, we take the same parameters as in the previous experiment, with $T = 4, 8, \ldots, 128$ and $N = 256, 512, \ldots, 8192$, and compute $G_1$ by the direct method. Then, for $\varepsilon = 10^{-4}$, we measure directly the same $\varepsilon$-ranks which were computed on the fly and are shown in Figs. 8a and 8c; that is, we simply compute the SVD of the blocks and count the number of singular values above $\varepsilon$. The results are given in Figs. 8b and 8d.

Comparing the online and offline results shows that our on the fly procedure in fact obtains smaller ranks than those computed offline from the full solution. This observation merits some comment. One would not expect the online and offline $\varepsilon$-ranks to be identical, since the history sums used in the online algorithm contain an error of magnitude $\varepsilon$. Our only concern would be if the smaller online $\varepsilon$-ranks were accompanied by an error of magnitude much larger than the expected $\varepsilon$, and this is not the case for any of the examples treated in Figs. 8b and 8d. More generally, a difference in the $\varepsilon$-ranks of two matrices does not imply a large difference between the matrices themselves, measured in some norm; for example, the $n \times n$ diagonal matrices $A = \operatorname{diag}(1, 1.5\varepsilon, \cdots, 1.5\varepsilon)$ and $B = \operatorname{diag}(1, 0.5\varepsilon, \cdots, 0.5\varepsilon)$ have $\varepsilon$-ranks $n$ and 1, respectively, and $\|A - B\|_2 / \|A\|_2 = \varepsilon$. In our examples, compared with the size of the matrices which have been compressed, the observed discrepancy in the ranks is in practice negligible.

We can now estimate the effectiveness of our method for other systems by this offline procedure. As an example, we use the Hubbard model, a paradigmatic problem in the theory of

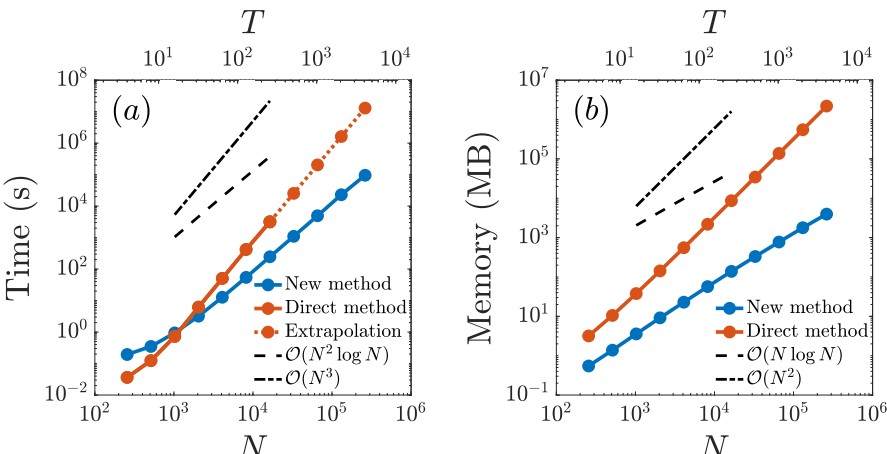

Figure 7: Increasing the propagation time $T$ with $N$ fixed time steps of size $\Delta t$ and SVD truncation tolerance $\varepsilon = 10^{-4}$, for the Floquet example. The plots are analogous to those shown for the ramp example in Fig. 1.

strongly correlated systems which demonstrates a variety of phenomena, including the metal-insulator transition and magnetic phases [86–89]. The Hamiltonian,

$$H(t) = -J \sum_{\langle i,j \rangle, \sigma} \exp^{-i\phi(t)} c_{i\sigma}^{\dagger} c_{j\sigma} + U \sum_i (n_{i\uparrow} - 1/2)(n_{i\downarrow} - 1/2), \tag{19}$$

describes the competition between the kinetic energy and the on-site Coulomb interaction. Here, $c_{i\sigma}$ is the annihilation operator at site $i$ for spin $\sigma$, $n_{i\sigma}$ is the corresponding density operator, $J$ is the hopping parameter, and $U$ is the Coulomb strength. Coupling to an external electric field $E(t)$ is introduced via the Peierls substitution and enters as a time-dependent phase of the hopping parameter $\phi(t) = -lA(t)$, where $l$ is the lattice constant. The vector potential $A$ is obtained from the electric field by $A(t) = -\int_0^t d\bar{t} E(\bar{t})$. We consider two characteristic cases at half-filling: the weak coupling regime, in which the bandwidth $W$ is larger the Coulomb interaction, $W > U$, and the system is metallic, and the strong coupling regime, in which the Coulomb interaction dominates, $U > W$, and the system is a Mott insulator.

**Correlated metal within GW** In the weak coupling regime, we consider the GW approximation for the self energy. This approximation has been used extensively for the realistic modeling of molecules, weakly correlated extended systems, coupling with bosonic excitations, and screening [32–36, 42, 43, 90, 91]. In combination with DMFT, it was used in and out of equilibrium to study plasmonic physics in correlated systems [30, 44, 92–95]. We consider a one dimensional setup with translational invariance and a paramagnetic phase; see Ref. [22] for a detailed description. In this case, the single particle energy includes the coupling to the external vector potential as $\epsilon(k - A) = -2J \cos(k - A)$.

We examine two excitation protocols. The first involves a short electric field pulse, as typically used in pump-probe experiments, parametrized by

$$E(t) = E_0 \sin(\omega(t - t_0)) \exp\left(-4.2(t - t_0)/t_0^2\right), \tag{20}$$

where the delay $t_0 = 2\pi/\omega$ is chosen so that the pulse contains one cycle. We use a pump strength $E_0 = 5$ and base frequency $\omega = 4$. The second is a Floquet driving of the electric field,

$$E(t) = E_0^F \sin(\omega_F t), \tag{21}$$

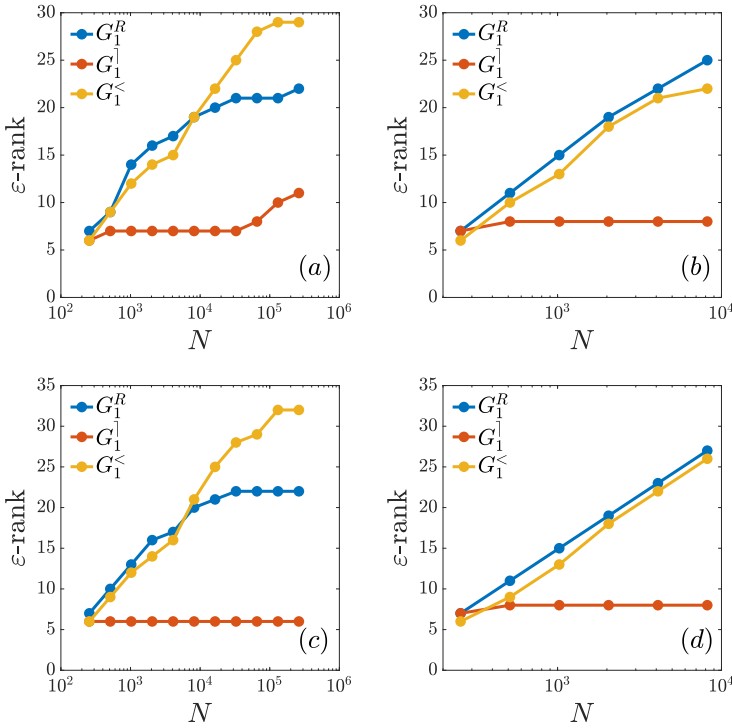

Figure 8: Ranks of the compressed representations for $G_1$ with increasing propagation times $T$ corresponding to $N$ time steps of fixed size $\Delta t$, and SVD truncation tolerance $\varepsilon = 10^{-4}$, for the ramp (first row, (a) & (b)), and Floquet (second row, (c) & (d)) examples. The first column ((a) & (c)) contains ranks computed on the fly, and the second ((b) & (d)) contains ranks computed offline for the smaller values of $N$.

with driving strength $E_0^F = 1$ and frequency $\omega_F = 2$. In both cases we fix the Coulomb strength $U = 2$, and at equilibrium the inverse temperature $\beta = 20$.

The time evolution of the kinetic energy $E_{\text{kin}}(t) = \frac{1}{N} \sum_k \epsilon(k) \langle c_k^\dagger c_k \rangle(t)$ is shown for the pulse excitation and the periodic driving in Figs. 9d and 9e, respectively. In the pulse excitation, the kinetic energy is transiently enhanced during the pulse and then quickly approaches the long-time limit which, as we are considering a closed system, is higher than the equilibrium kinetic energy. In the periodically driven case, the kinetic energy gradually grows toward the expected infinite temperature state $E_{\text{kin}} = 0$ as the system heats up. We note that for readability, $E_{\text{kin}}(t)$ is plotted on a much shorter time interval than that of our longer simulations.

We use the NESSi library [22] to solve the time-dependent GW equations for these systems. We fix the time step $\Delta t = 0.01$ and the Matsubara time step $\Delta \tau = 0.04$, and compute solutions $G_k(t, t')$ for $T = 3.75, 7.5, \ldots, 60$, corresponding to $N = 250, 500, \ldots, 4000$, for both the pulse and Floquet examples. We then measure the $\varepsilon$-ranks of all blocks in the compressed representation for $\varepsilon = 10^{-4}$, and use these values, along with the number of directly stored matrix entries, to compute the total memory required to store each Green's function in the compressed representation. The results are shown in Fig. 9. The ranks grow slowly with $N$. Even for $N = 4000$, we observe a compression factor of over 30 for the pulse example and 20 for the Floquet example; this can be compared with a compression factor of approximately 25 for the Falicov-Kimball examples with $N = 4096$. Of course, because of the near linear scaling of the memory usage, the compression factors will increase nearly linearly with $N$. As

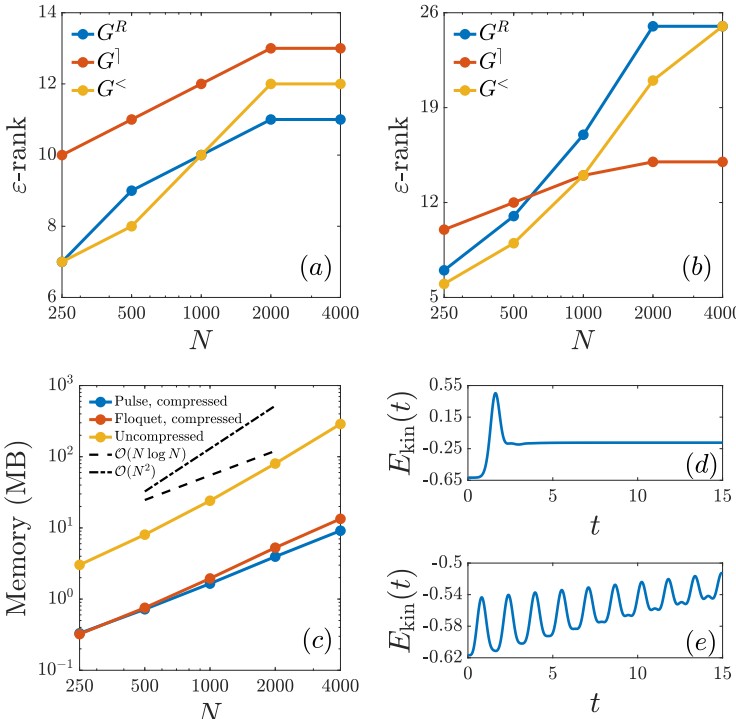

Figure 9: Ranks and memory usage for the GW examples with increasing propagation times $T$ corresponding to $N$ time steps of fixed size $\Delta t$, computed offline from solutions obtained using the NESSi library. Panels (a) and (b) give $\varepsilon$-ranks for the pulse and ramp examples, respectively, with $\varepsilon = 10^{-4}$. Panel (c) gives the total memory usage in each case using compressed and direct storage. Panels (d) and (e) show the time evolution of the kinetic energy for the pump and Floquet examples, respectively.

the two excitations represent very different physical regimes, this experiment gives evidence of the broad applicability of the HODLR compression technique.

**Mott insulator within DMFT**    We next treat a strongly correlated Mott insulator. The description of the Mott insulating phase of the Hubbard model requires a nonperturbative approach, and we use the time-dependent DMFT description [17]. For simplicity, we consider the Bethe lattice self-consistency condition

$$\Delta(t, t') = J(t)G_{\text{loc}}(t, t')J(t'). \tag{22}$$

Here we have introduced the hybridization function $\Delta$ and the local Green's function $G_{\text{loc}} = G_{ii}$. In the DMFT description, the lattice problem is mapped to an effective impurity problem, and we use the strong coupling expansion NCA as the impurity solver; see Ref. [41] for details. To describe the electric field on the Bethe lattice, we have followed the prescription in Refs. [96–98].

As in the weak coupling case, the first excitation protocol is a short electric field pulse of the form (20) with a single cycle. We use $E_0 = 5$ and $\omega = 5$. The second is a periodic driving of the form (21), with $E_0^F = 1$ and $\omega_F = 5$. In both cases we set $U = 6$ and $\beta = 20$.

The kinetic energy in the DMFT description is given by $E_{\text{kin}}(t) = -2i(\Delta * G_{\text{loc}})^<(t, t)$. During the pulse the kinetic energy, shown in Fig. 10d, increases and then quickly relaxes to the long time limit. Despite the rather fast relaxation to a nearly constant value of the kinetic energy, computing this result is far from trivial as it requires integration over several

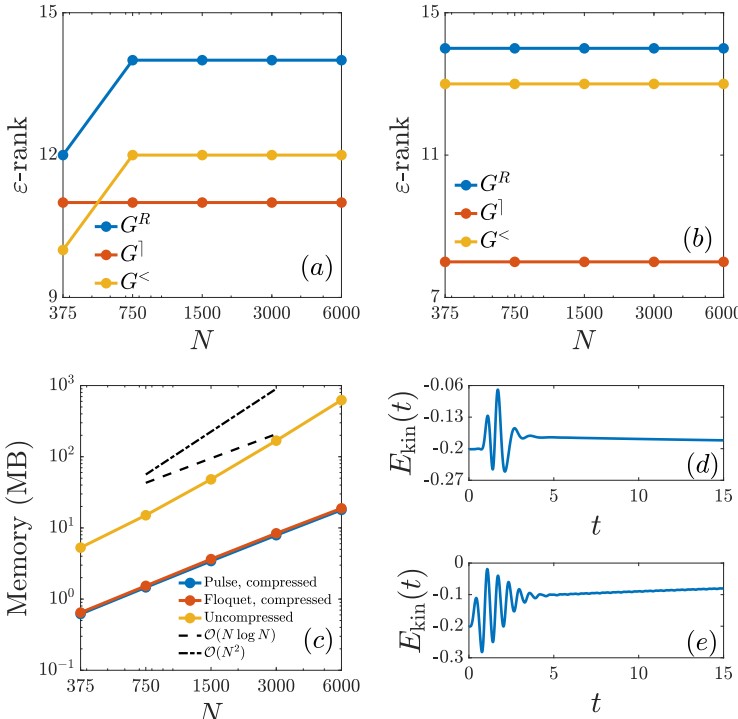

Figure 10: Ranks and memory usage for the DMFT examples with increasing propagation times $T$ corresponding to $N$ time steps of fixed size $\Delta t$, computed offline from solutions obtained using the NESSi library. The plots are analogous to those shown for the GW examples in Fig. 9.

highly oscillatory auxiliary functions, often called pseudo-particle propagators; see Ref. [41] for details. In the Floquet example, the kinetic energy approaches the infinite temperature state $E_{\text{kin}} = 0$ in the long-time limit. While the initial dynamics show strong oscillations, these are rapidly damped in the resonant regime; see Refs. [45, 46, 78].

We fix the time step $\Delta t = 0.02$ and the Matsubara time step $\Delta \tau = 0.04$, and compute solutions $G_k(t, t')$ for $T = 7.5, 15, \ldots, 120$, corresponding to $N = 375, 750, \ldots, 6000$, for both the pulse and Floquet examples. We then measure $\varepsilon$-ranks for $\varepsilon = 10^{-4}$ and memory usage as in the GW examples. The results are given in Fig. 10. The ranks remain constant as $N$ increases, and the memory scaling is consequently ideal. For $N = 6000$, we obtain compression factors of over 30 for both examples. Moreover, we have confirmed that a similar degree of compression may be obtained for the auxiliary pseudo-particle propagators. We note that in these examples, the Green's functions and self energies exhibit rapid decay in the off-diagonal direction, consistent with the observed rank behavior, so both our method and methods based on sparsity are applicable [50, 51]. This demonstrates, in addition, that matrices with rapid off-diagonal decay are in particular HODLR compressible.

# 5 Conclusion

We have presented a numerical method to reduce the computational and memory complexity of solving the nonequilibrium Dyson equation by taking advantage of the HODLR structures of Green's functions and self energies observed in many physical systems. The method works by building TSVD-based compressed representations of these functions on the fly, and using them

to reduce the cost of evaluating history integrals. We have confirmed significant compressibility for various models of interest, including instances of the Falicov-Kimball and Hubbard models in different diagrammatic approximations. The accuracy of our method, compared with direct time stepping methods, is controlled by the user, and in particular does not involve any new modeling assumptions. Selection of compression parameters is automatic, so our method may be used as a black box on new systems with unknown structure.

This work suggests many important topics for future algorithms research, of which we mention a few.

- Our method is compatible with more sophisticated discretization techniques, like high-order time stepping [22]. While these do not by themselves change the computational and memory complexity of the solver, they yield a significant reduction in the constant associated with the scaling, and should be used in implementations. Also, in the equilibrium case, spectral methods and specialized basis representations have been used to represent Green's functions with excellent efficiency, and their applicability in the nonequilibrium case has not yet been explored [61, 99–102].

- A distinction must be made between automatic compression methods, like the one we have described, and adaptive discretizations, which adjust grids to increase resolution in certain regions of the solution. Though significant technical challenges remain, combining compression and high-order discretizations with automatically adaptive time stepping would enable the simulation of much more sophisticated systems at longer propagation times, and we envision such methods becoming the standard in the long term.

- For systems amenable to HODLR compression, it remains to determine whether this structure can be used to reduce other bottlenecks. In particular, the evaluation of high-order self energy diagrams involves a sequence of nested convolutions with potentially structured operators.

- The effectiveness of HODLR compression in solving the nonequilibrium Dyson equation is unsurprising, as various forms of hiearchical low rank compression are commonly used in scientific computing to compress integral operators with kernels that are smooth in the far field. However, since the equations are nonlinear, the degree of compressibility is difficult to analyze, and it remains to determine the limits of our approach. If HODLR compression is not applicable to some systems, it may still be possible to use similar ideas with other compression techniques from the numerical linear algebra and applied mathematics literature. Indeed, significant progress has been made over the last several decades on exploiting various types of data sparsity, especially in the context of partial differential equations and associated integral equations, and an opportunity remains to apply these techniques to Green's function methods.

A full implementation of our algorithm in the high-order time stepping code NESSI [22] is forthcoming, and will be reported on at a later date.

## Acknowledgements

Simulations for the Hubbard model were carried out on the Rusty cluster at the Flatiron Institute. For the solution of the strong coupling Hubbard model, we have used a numerical library developed by Hugo U. R. Strand and Martin Eckstein. We would like to acknowledge useful discussions with Martin Eckstein, Yuta Murakami, Michael Schüler, and Alex Barnett. The Flatiron Institute is a division of the Simons Foundation. D.G. acknowledges the support of the Slovenian Research Agency (ARRS) under Programs No. J1- 2455 and P1-0044.

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
