# Peer review of "Low rank compression in the numerical solution of the nonequilibrium Dyson equation"

_SciPost Physics, doi:SciPost Phys. 10, 091 (2021)_

## Round 2 · Referee Report · Anonymous (Referee 1) · 2020-12-16

Strengths

  1. Very interesting research avenue and methodology.
  2. Very careful study, clearly and precisely written.

Weaknesses

  1. None really, although the actual applicability of the approach to new interesting problems remains to be further established

Report

This article proposes an algorithm to speed up the calculation of Keldysh Green’s functions from a given approximation scheme of the self energy. Its chief results are - The empirical observation that the Green’s function maybe compressed and encoded with a set of truncated singular values decomposition of its sub-blocks. They call the corresponding structure hierarchical off- diagonal low rank (HODLR) property. - They use HODLR to construct an algorithm for fast integration of the Green’s functions equation of motions including the integral memory kernel terms. - They benchmark the algorithm on a Falicov-Kimball model and further show that two other models have the HODLR structure.

This is a very neat paper with an original creative idea. The manuscript itself is very clear and well written and the benchmarks convincing. The results may have direct applications to a number of situations and may also stir new developments for other approaches. I fully favor publication.

I have only three questions/comments.

  • I think it would be better to merge Figure 8 and 9 so that we can more easily compare the inline rank (calculated within the calculation) with the offline rank (computed after the fact as an external check). It is an important aspect of the method to check that the inline rank can be trusted.

  • As a matter of fact, it seems that the inline calculation of the rank of G^R in Fig.8b saturates to 22 while the « real » offline rank in Fig. 9b grows up linearly to at least 26 with no sign of saturation. Could the authors comment of this? It could indicate that the method would fail silently at long time. More precisely, I am concerned with the accumulation of errors in the scheme for the update of the blocks svd decomposition as I have often witnessed that these low rank update algorithms are often numerically unstable.

  • It would be interesting to have more insights as to what is at the origin of the HODLR structure. Is it related to some physical property? The authors quickly discuss it in the conclusion but very shortly.

Requested changes

None, but the authors may consider the comments at the end of the report.

  • validity: top
  • significance: high
  • originality: high
  • clarity: top
  • formatting: excellent
  • grammar: excellent

Author:  Jason Kaye  on 2020-12-18  [id 1092]

(in reply to Report 1 on 2020-12-16)

We thank the referee for their useful remarks. Below are our responses to each of the three comments.

(1) We agree that this comparison is important, and will merge the figures.

(2) It is indeed interesting that the ranks computed online are slightly lower than those computed offline. However, we do not believe this is a cause for concern, for the following reasons.

(i) It is certainly possible to have two matrices which are within epsilon of each other in the 2-norm or entrywise, but for which the epsilon-ranks are arbitrarily far apart. Consider, for example, the NxN diagonal matrices with first entry 1 and remaining nonzero entries 1.1*epsilon and 0.9*epsilon, respectively. Their difference is 0.2*epsilon in the 2-norm and entrywise, but the first has full epsilon-rank, whereas the second has epsilon-rank 1.

(ii) The important question, then, is whether or not the error between the solution computed using the direct method and our method is greater than epsilon in the cases that the ranks differ. For the examples shown in Fig 9 (up to N = 8192), we have verified that the errors are below epsilon as expected, even though we have entered the regime in which the ranks differ. In fact, the largest entrywise error consistently occurs for a rather small time step index, not in the large time regime, so longer propagation does not increase the entrywise error at all.

There is no indication that this difference in ranks represents a long-time accumulation of error or numerical instability. We have observed in every example we have tried that the error tolerance epsilon is achieved, and have never observed a numerical instability. Furthermore, one would typically expect larger, not smaller, ranks to appear if the rank-revealing algorithm were unstable, or if a numerical instability were amplifying noise created by our SVD truncation of the history contribution to the integral equation. Our observation is more consistent with damping rather than amplification of truncation errors. Of course, we have not offered proof that the method is unconditionally stable, and cannot guarantee that instability is impossible, but in the present nonlinear setting proving even that the direct method is stable would be rather nontrivial.

(iii) The observation is in fact not so surprising. One would not expect a priori that the epsilon-ranks computed online and offline are exactly the same, since the history used in the online algorithm is only approximate; only, at worst, that a stable time stepping method should not increase the ranks significantly. The observed difference in ranks, compared to the size of the full matrix, is negligible, and unimportant in practice - for the ramp example, it is 19 compared with 25 for a matrix of size 4096x4096.

While this is an interesting phenomenon and well worth future investigation, its mechanism may be rather subtle. For the reasons listed above, we do not believe it indicates a likely failure mode, at least for the examples presented. We will add some clarifying remarks to the text addressing this issue.

(3) Investigating the physical origin of the HODLR structure from an empirical and an analytical perspective will certainly be a topic of our future research. The HODLR structure is commonly found in the context of compact integral operators, and a low-rank ansatz is routinely used to compress rather generic smooth functions. Since the Volterra integral operators arising in the Dyson equation typically have piecewise smooth kernels and are therefore compact, the HODLR structure is a natural one to try. Nevertheless, we are considering a fairly general class of nonlinear equations, so a full analysis is not expected to be straightforward. We will add some comments discussing these issues.

We should emphasize that other compression methods exist, including more sophisticated methods based on low-rank structure like H matrix compression, and others, like the butterfly compression method, designed for more general oscillatory operators. We hope that this work will be the first of many which investigate linear algebraic compression strategies for this class of problems, and while we expect the HODLR structure to be quite broadly applicable, it would not be surprising if distinct yet related methods are required in some cases.

---

## Round 2 · Referee Report · Anonymous (Referee 2) · 2021-1-11

Strengths

1-New numerical methodology with a high potential for new computational discoveries in the field of strongly correlated systems;
2-Application to two paradigmatic models in dynamical mean-field theory show an impressive order-N improvement in both run time and memory usage;
3-Solid research, clearly reported;
4-Several venues of future research identified.

Weaknesses

1-None.

Report

In this paper, Jason Kaye and Denis Golez propose an efficient low-rank compression scheme for solving Green’s functions within the Keldysh formalism of nonequilibrium field theory. Their scheme overcomes what is arguably the major obstacle in the numerical solution of nonequilibrium Dyson equations (i.e. the evaluation of history integrals) by means of a compressed representation of off-diagonal blocks of Green’s functions with control over target accuracy. The novel approach is applied to representative models with different driving protocols and thoroughly compared to a standard direct time stepping method. Their results show that the low-rank compression scheme allows unprecedented large propagation times with modest computational resources, reflecting a massive improvement both in the algorithm complexity (from N^3 to N^2 Log N) and memory usage (from N^2 to N Log N). In my view, this is a breakthrough for the numerical study of strongly correlated systems and could trigger interesting developments in problems deemed intractable by previous methods. The planned implementation in the open-source NESSI package already hints at the relevance of the new low-rank compression scheme developed by the authors. The paper is clearly written and well referenced, and I’m happy to recommend publication in its current form.

Requested changes

N.A. (see report)

---

## Round 3 · List of Changes

We thank the referees for their useful comments. In response, we have made the following revisions.

  • Figs. 8 and 9 have been merged, and the surrounding discussion has been modified accordingly.

  • We have added a discussion on pg. 10 concerning the discrepancy in online and offline ranks.

  • We have added a remark in the fourth bullet point in the conclusion addressing the origin and naturalness of the HODLR compressibility.

---

## Editorial Decision

published